# Protocol for the SAFEST review: the Shock-Absorbing Flooring Effectiveness SysTematic review including older adults and staff in hospitals and care homes

Amy Drahota ,[1] Lambert M Felix ,[1] Bethany E Keenan ,[2] Chantelle C Lachance ,[1] Andrew Laing ,[3] Dawn C Mackey ,[4] James Raftery [5]

For numbered affiliations see end of article.

**Correspondence to**
Dr Amy Drahota;
amy.drahota@port.ac.uk

## ABSTRACT

**Introduction** Falls in hospitals and care homes are a major issue of international concern. Inpatient falls are the most commonly reported safety incident in the UK's National Health Service (NHS), costing the NHS £630 million a year. Injurious falls are particularly life-limiting and costly. There is a growing body of evidence on shock-absorbing flooring for fall-related injury prevention; however, no systematic review exists to inform practice.

**Methods and analysis** We will systematically identify, appraise and summarise studies investigating the clinical and cost-effectiveness, and experiences of shock-absorbing flooring in hospitals and care homes. Our search will build on an extensive search conducted by a scoping review (inception to May 2016). We will search electronic databases (AgeLine, CINAHL, MEDLINE, NHS Economic Evaluation Database, Scopus and Web of Science; May 2016–present), trial registries and grey literature. We will conduct backward and forward citation searches of included studies, and liaise with study researchers. We will evaluate the influence of floors on fall-related injuries, falls and staff work-related injuries through randomised and non-randomised studies, consider economic and qualitative evidence, and implementation factors. We will consider risk of bias, assess heterogeneity and explore potential effect modifiers via subgroup analyses and sensitivity analyses. Where appropriate we will combine studies through meta-analysis. We will use the GRADE (Grading of Recommendations, Assessment, Development and Evaluations) approach to evaluate the quality of evidence and present the results using summary of findings tables, and adhere to the Preferred Reporting Items for Systematic Reviews and Meta-Analyses reporting guidelines.

**Ethics and dissemination** We will follow the ethical principles of systematic review conduct, by attending to publication ethics, transparency and rigour. Our dissemination plan includes peer-reviewed publication, presentations, press release, stakeholder symposium, patient video and targeted knowledge-to-action reports. This review will inform decision-making around falls management in care settings and identify important directions for future research.

**PROSPERO registration number** CRD42019118834.

## Strengths and limitations of this study

► This will be a mixed methods systematic review including randomised and non-randomised clinical studies, economic and qualitative evidence.
► Studies will be assessed using the updated Cochrane risk of bias tools for quantitative evidence, and Joanna Briggs Institute method for qualitative studies.
► Analyses will be at the study level, which limits the scope for exploring moderating factors related to patient-level characteristics on the effectiveness of flooring interventions.
► The quality of the evidence will be summarised using the GRADE approach, with the strength of the review's findings limited to the quantity and internal validity of the included studies.
► We will be guided by the Knowledge-to-Action Framework to facilitate the translation of the findings into practice.

## INTRODUCTION

Falls in health and social care settings are a major concern for older adults globally, causing morbidity, mortality and economic burden.[1–3] Falls have been climbing the league tables of the leading causes of global disability-adjusted life years,[4] with falls and injury rates in residential care settings substantially higher than that of older people living in the community.[1] In the UK, inpatient falls are the most commonly reported safety incident with over 250 000 reported per year in the National Health Service (NHS) in England alone.[5] Falls have a complex aetiology of

BMJ

intrinsic (eg, co-morbidities, cognitive function, mobility) and extrinsic (eg, environmental design, staffing, footwear, medication) risk factors[3 6–8] and no single solution effectively prevents them. A systematic review of falls prevention interventions in institutional settings,[9] found low quality evidence, with uncertain conclusions for a range of interventions including: exercise, physiotherapy, sensor alarms and multifactorial interventions. This review excluded studies targeting fall-related injury prevention, yet the prevention of severe falls is considered a priority.[5] One of the most severe consequences of falls are hip fractures, but wearable hip protectors have poor compliance which is a barrier to their use.[10] Unlike hip protectors, manipulating the physical environment is a promising intervention for reducing injurious falls as it requires no compliance from patients or staff, and can accommodate other injury types.

Shock-absorbing flooring can reduce the impact forces of falls by decreasing the stiffness of the ground surface.[11] However, softer floors could negatively impact on gait, potentially leading to increased falls risk.[11–15] The potential benefits and risks of shock-absorbing floors may vary depending on the type of patient utilising them. Furthermore, adverse effects of shock-absorbing floors may present in staff if greater effort is required to manoeuvre rolling equipment, potentially increasing injury risk.[16]

There has been no comprehensive systematic review focusing on flooring interventions in healthcare settings for fall-related injury prevention. A recent scoping review of flooring interventions involved a thorough search of the literature; however, it did not involve a critical appraisal or systematic synthesis.[17] A systematic review of studies identified in the scoping review[16 18–25] as well as more recent studies,[26–31] will provide a more reliable basis for decision-making and identify the next steps for research.

This publication is an abridged version of the full protocol,[32] and is registered on PROSPERO.[33] Any important protocol amendments will be published on these platforms.[32 33] We have conformed to the Preferred Reporting Items for Systematic Reviews and Meta-Analyses Protocols (PRISMA-P)[34] in the writing of this protocol (online supplementary file 1).

## Aims and objectives
We aim to systematically review the evidence on shock-absorbing flooring use in care settings (hospitals and care homes) for fall-related injury prevention in older adults. Specifically, we will:
1. Assess the potential benefits (fall-related injury prevention) and risks (falls; staff injuries) of different flooring systems in care settings.
2. Assess the extent to which these potential benefits and risks may be modified by different study/setting, intervention and participant characteristics.
3. Critically appraise and summarise current evidence on the resource use, costs and cost-effectiveness of shock-

absorbing flooring in care settings for older adults, compared with standard flooring.
4. Summarise findings on the implementation of flooring interventions in the included studies.
5. Summarise the views and experiences of shock-absorbing flooring use from staff, patients'/residents' and visitors' perspectives.
6. Identify gaps in existing evidence.

## METHODS
### Eligibility criteria
#### Population
The target population for the intervention to potentially benefit is older adults in care settings. We have no set cut-off criteria for age, as chronological age may not be a good indicator of frailty.[35] Studies must focus on adult populations to be included; studies focusing solely on children will be excluded. We are also interested in staff outcomes.

#### Setting
Studies must have been conducted in a care setting (defined below) including hospitals (acute, sub-acute), intermediate and long-term care settings (nursing and care homes). Studies conducted in people's own homes, or other settings (eg, playgrounds, sporting venues) will be excluded.

Care settings will be broadly defined as[36]:
► Care home environments (a facility that provides: communal living facilities for long-term care; overnight accommodation; nursing or personal care; for people with illness, disability or dependence).
► Hospital environments (a facility that provides: communal care where there is an expectation that this care is time limited; overnight accommodation; nursing and personal care for people with illness and disability).

#### Interventions
Interventions may include flooring systems which have been purposely designed to prevent fall-related injuries (eg, SmartCells, Sorbashock, Kradal), thick vinyl (>5 mm thick; eg, sports floors, such as Tarkett Omnisports Excel), carpet with or without underlay, and other combination flooring systems (eg, vinyl overlays with padded underlays, such as foam or rubber, or wooden subfloors). Alternative terminology for the intervention may include variations on the terms: compliant flooring, safety flooring, soft flooring, impact absorbing flooring, energy absorbing flooring, low-impact flooring, dual stiffness flooring, low stiffness flooring, absorptive surfaces, cushioned flooring, rubber flooring, acoustic flooring and carpet.

We will exclude studies reporting exclusively on mats as they are not permanently affixed to the floor and do not provide universal coverage or protection; mats have different implications for installation and practice and are not the focus of this review. Studies in which flooring is one component of a package of interventions and the

effects of the floor cannot be distinguished from concurrent interventions will be excluded.

## Comparator

Our main control group is standard or rigid flooring (eg, concrete subfloor,≤2 mm vinyl/resilient flooring). We will include head-to-head comparisons of different types of shock-absorbing flooring systems where possible. Studies may compare any combination of flooring systems (subfloors and overlays).

## Outcomes

The reporting of specific outcomes does not form part of our eligibility criteria for studies to be included in this review.

## Study design

We will include randomised, non-randomised, observational, economic and qualitative studies. While randomised trials of flooring are feasible, the nature and logistics of the intervention make observational and opportunistic quasi-experimental designs more practical. Studies will be classified according to their component design features using the study design features presented in the Cochrane Handbook.[37] The following study designs will be eligible:

► Individually or cluster randomised controlled trials.
► Quasi experimental studies where allocation is non-random.
► Interrupted times series.
► Controlled before and after studies.
► Cohort studies.
► Case-control studies.
► Partial and full economic evaluations, based on a single study or model.
► Qualitative studies to explore experiences, attitudes and perceptions towards flooring interventions.

We will exclude simple before and after studies measuring quantitative outcomes, with no evaluation of time trends or concurrent control.

## Information sources and search strategy

To avoid duplication of effort, we will build on the search already conducted in a scoping review[17] which completed its search in May 2016. The clinical[12 16 18–26 38–46] and cost-effectiveness[11 21 26 40 41 47–63] records identified in the scoping review will be assessed for inclusion in the current review. We will continue the search from May 2016 to present, and will not apply any language restrictions. A comprehensive search, as listed in table 1, will be undertaken, to include electronic databases, grey literature, hand searches, citation screening and expert consultation.

We have adapted the broader search strategy of the scoping review[17] to make it more specific to the current study (The SAFEST Review). The strategy for MEDLINE (online supplementary file 2) is based on our eligibility criteria and uses a combination of keyword synonyms and controlled vocabulary terms (eg, MeSH). We will adapt the MEDLINE search for other information sources.

## Study records
### Data management

We will import the search records into EndNote online and use Covidence[54] to support duplicate record identification, screening, data collection and risk of bias assessment processes, identification and resolution of discrepancies, and producing a PRISMA flow diagram.[55] Data analyses will be undertaken in RevMan,[56] and summary of findings (SoF) tables and Evidence Profiles will be created using GRADE Pro.[57 58]

### Selection process

We will screen titles, abstracts and full reports independently in duplicate using an eligibility checklist. All records included in the clinical and cost-effectiveness sections of the scoping review will be assessed at the full report stage. From the results of the updated search, we will begin by screening titles, and those that look potentially relevant will be reviewed in abstract form. We will then screen the full texts of records that appear definitely or possibly relevant. Discrepancies will be resolved through a third independent arbitrator.

### Data collection process

Our theoretical framework of potential effect modifiers (figure 1) will underpin the data collection process. We will develop and pilot the data collection form with a data collection manual. Two reviewers will independently undertake data collection and assessment of risk of bias.

Data collection will include the following key components of information:

► Study identification.
► Time/duration and geographical place of conduct.
► Participant characteristics.
► Intervention(s).
► Control(s).
► Outcome data acquisition: method of falls reporting; classification system of injuries; identification of fractures (confirmation of diagnosis/type of fractures included); identification of adverse effects.
► Setting.
► Study design.
► Risk of bias.
► Outcomes data.
► Patient and public involvement in the research.
► Follow-up questions for study authors.

## Outcomes and prioritisation

There is no core outcome set specifically for flooring interventions; however we have considered the common outcome data set for fall injury prevention trials in community-dwelling populations[59] and the international consensus statement for trials on hip protectors.[60] Recognising the unique features of our review and through stakeholder engagement[61] and discussion with our public

**Table 1** List of information sources

| Search type | Information sources |
| --- | --- |
| **Electronic databases** | AgeLine (EBSCO)<br>CINAHL Complete (EBSCO)<br>MEDLINE (EBSCO)<br>NHS Economic Evaluation Database (Centre for Reviews and Dissemination)<br>Scopus<br>Web of Science (Thomson Reuters) |
| **Grey literature search** | **Clinical trial registries**<br>WHO International Clinical Trials Registry Platform<br>**Theses/dissertations**<br>ProQuest Theses and Dissertations<br>**Abstracts/conference proceedings**<br>Biennial Conference of the Australian and New Zealand Falls Prevention Society<br>Canadian Association on Gerontology Annual Scientific and Educational Meeting Gerontological<br>Society of America's Annual Scientific Meeting<br>International Society for Posture and Gait Research World Congress<br>World Conference of Gerontechnology<br>World Congress of the International Association of Gerontology and Geriatrics<br>**Websites**<br>Agency for Healthcare Research and Quality<br>Canadian Agency for Drugs and Technologies in Health<br>NHS Improvement<br>NICE Guidelines<br>Open Grey (opengrey.eu)<br>Parachute Canada<br>The National Institute for Occupational Safety and Health<br>The International Network of Agencies for Health Technology Assessment<br>UK Health Technology Assessment<br>US Center for Health Design<br>WHO Health Evidence Database |
| **Hand searching and citation screening** | **Reference lists**<br>References of included studies<br>Forward citation searching of included studies in Web of Science<br>**Journal**<br>Age and Ageing |

NHS, National Health Service; NICE, National Institute for Health and Care Excellence.

involvement group, we have prioritised the following outcome measures:

Primary outcomes:
► Injurious falls rate per 1000 person-bed days.
► Falls rate per 1000 person-bed days.

These measures assess the potential benefits and harms of flooring interventions for patients/residents, accounting for bed occupancy levels and follow-up time; injurious falls rate additionally accounts for variations to the underlying falls rate, as a pragmatic measure of effectiveness.

Secondary outcomes:
► Number of falls with injuries (eg, none, minor, moderate, severe, death).
► Number of fractures.
► Number of hip fractures.
► Number of fallers (risk of falling ≥1 times).
► Number of adverse events (eg, staff injuries as defined by study authors).
► Number of head injuries.

► Fractures per 1000 patient-bed days.
► Hip fractures per 1000 patient-bed days.
► Qualitative outcomes (eg, attitudes, views, and experiences of staff, patients/residents, visitors).
► Economic outcomes (to include assessments of quality-adjusted life years).
► Process outcomes (eg, ease of, or problems with, flooring installation).

**Risk of bias in individual studies**

Risk of bias assessment will be undertaken using the updated Cochrane risk of bias tool (ROB 2.0) for randomised trials[62] and the Risk Of Bias In Non-randomised Studies of Interventions (ROBINS-I) tool[63] for other quantitative designs. We will assess the risk of bias at the level of study results. Review authors will not be blinded during risk of bias assessments; however, where they have been involved in co-authoring an included study, assessments will be undertaken by at least two other independent reviewers. Supporting information and

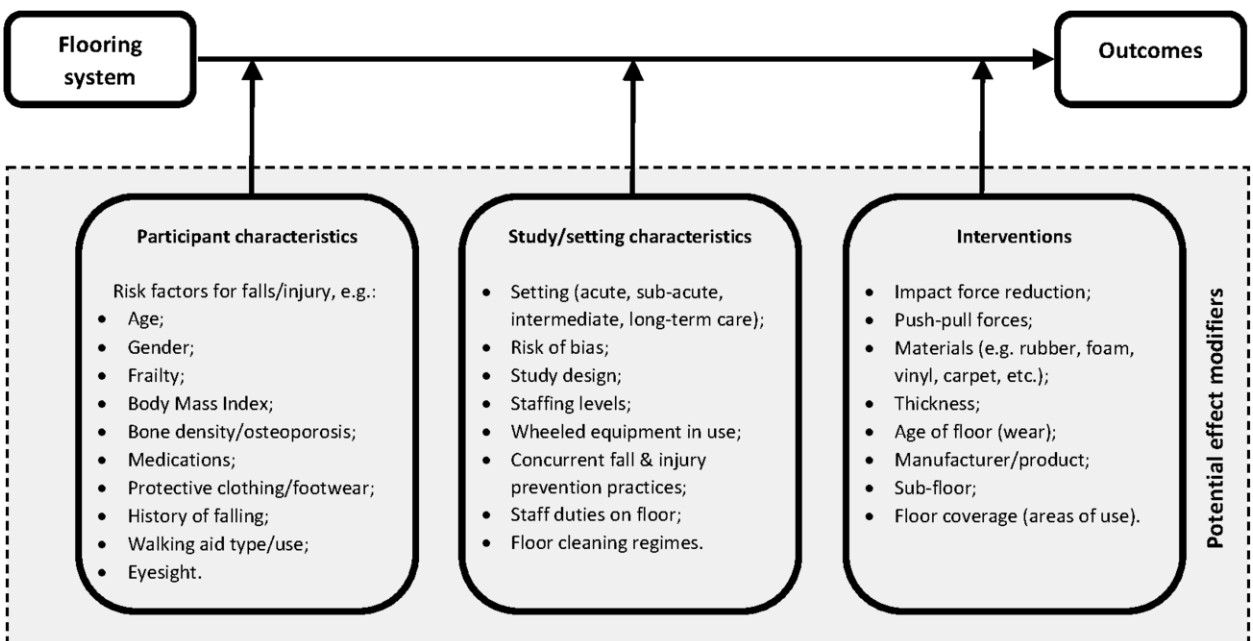

**Figure 1** Theoretical framework of potential effect modifiers.

justification for judgements (high, low, some concerns) will be recorded for each bias domain. We will follow the guidance to derive summary judgements for each outcome, which will be used to inform our sensitivity analyses and GRADE assessments.[57]

### Data analysis (quantitative studies)
#### Dealing with missing data
We will seek further information from study authors where required. If missing data are from participant/cluster dropouts, analyses will be based on the available data and an assessment of the problem will be included as part of our risk of bias judgements.

#### Measures of treatment effect
We will report rates of injurious falls, falls and fractures using incidence rate ratios and 95% CI. We will use risk ratios (95% CI) to describe number of fallers, number of falls with fall-related injuries and number of participants with fall-related fractures or head injuries. Where available we will also report hazard ratios for falls including all falls from recurrent fallers. For non-randomised studies, we will record the unadjusted and adjusted estimates and note the factors adjusted for. Where multiple adjusted estimates are presented, we will extract the estimate highlighted as the primary model by the authors, or where this is unclear, take the model which has adjusted for the most covariates. Where rate ratios or risk ratios are not reported, we will calculate them where feasible.[37]

Where studies present a break-down of the severity of injuries (as ordinal outcome data, eg, none, mild, moderate, severe, death), we will present these descriptively, and if studies have used similar categorisation systems, using figures where feasible. We will report adverse events to staff as a risk or rate ratio (per 100 working staff-days) where possible, or as the number of events observed during the follow-up period, if no clear denominator is known.

#### Unit of analysis issues
To avoid the issue of double counting, we will link multiple associated publications together. When primary studies include multiple study arms, we will either combine the groups (if logical) or include only one pair-wise comparison (intervention vs control) in any one analysis. In the case of cluster randomised trials we will take clustering into account, and plan to adjust the estimates using an intra-cluster correlation coefficient (ICC) borrowed from another similar study if required.[37]

#### Assessment of reporting bias
Where possible, we will produce funnel plots with different plotting symbols to identify subgroups. Funnel plot asymmetry will be tested if there are sufficient data (at least 10 studies to be combined), and visual inspection of the plots will be used to interpret the findings.

#### Data synthesis
Should meta-analysis be viable, studies will be combined using a random-effects model, assuming that intervention effects are likely to vary across studies (figure 1). We will use the generic inverse variance data type to produce forest plots in RevMan[56]; this method requires entering the natural logarithm of the rate ratio or risk ratio and its SE for each study. We will use 95% CIs throughout. Where evidence exists from randomised and non-randomised studies, we will report the data separately, giving more emphasis to the findings from randomised trials. We will organise non-randomised studies according to whether data collection was prospective or retrospective, and if controls were concurrent or historical. If appropriate, we will combine the data from randomised and

non-randomised studies to provide an overall summary effect estimate.

## Assessment of heterogeneity

Heterogeneity will be explored irrespective of whether we decide to pool studies in a meta-analysis. Heterogeneity will be assessed through a combination of visual inspection of the forest plots, along with consideration of tests for homogeneity ($\chi^2$ with statistical significance set at p<0.10), and measures for inconsistency ($I^2$) and heterogeneity ($tau^2$).

Where feasible, we plan to undertake subgroup analysis based on:
► Study design (randomised, non-randomised).
► Study setting (hospital, care home).
► Acuity of care (acute, sub-acute, intermediate, long-term care).
► Flooring type (novel shock-absorbing flooring, thick vinyl/vinyl & underlay, carpet, wooden subfloor).

## Sensitivity analyses

Sensitivity analyses will be undertaken based on:
► Risk of bias (eg, removing studies at high risk of bias on the ROB 2.0 tool, or critical/serious risk of bias on the ROBINS-I tool).
► Choice of effect estimates (eg, where multiple adjusted estimates are reported, the analysis will be run on the most optimistic and pessimistic scenarios).
► Adjustment for clustering where an ICC has been borrowed from another study (eg, we will assess the impact of opting for more or less conservative adjustments).

## Synthesis of qualitative studies

A meta-aggregative approach will be used to synthesise data from qualitative studies.[64] We will derive generalisable statements, in the form of recommendations that can be used to guide end-users of the review (eg, NHS chief executives, care home managers, estates/facilities managers, healthcare designers and builders, health and social care professionals, patients, residents and carers). Studies will be critically assessed using the Joanna Briggs Institute's critical appraisal tool.[64] We will follow the data collection process as above, and use QSR NVivo software for data analysis.[65]

## Synthesis of economic evidence

We will align our approach for the incorporation of costs data to an exemplar systematic review by Garrison and colleagues.[66] One reviewer (LF) will extract all data from included economic evaluations, which will be checked by an expert reviewer (JR). Our data extraction form will be based on the format and guidelines used to produce structured abstracts of full economic evaluations for inclusion in the NHS Economic Evaluation Database. The methodological quality of included economic evaluations will be assessed through the use of recognised checklists based on guidelines for economic submissions to the British Medical Journal (for economic evaluations based on a

single study),[67] and for quality assessment in economic decision-analytic models (for model-based economic evaluations).[68] Data extraction will include study characteristics such as country, settings, aims and methodological aspects related to economic evaluation, individual items within the respective checklists,[67 68] and the economic variables. We will collect the following economic variables, if reported: costs of flooring (purchasing, installation, maintenance); costs of falls based on injury, such as hospital resources (eg, increased length of stay, additional surgery needs), and post-discharge healthcare cost (eg, hospital readmission, outpatient visits); utility measures such as quality of life, life years and quality adjusted life years; and summary measures such as incremental cost-effectiveness ratio, net monetary benefits and value of information.

We will classify economic evaluations by type (*Partial evaluations*: 'outcome description', 'cost description', 'cost-outcome description', 'efficacy or effectiveness evaluation' or 'cost-analysis'; *Full economic evaluations*: 'cost-effectiveness analysis', 'cost-utility analysis' or 'cost-benefit analysis') and as either an economic evaluation based on a single study or a model-based economic evaluation. Where necessary, additional information from study authors will be sought.

Results will be tabulated and summarised narratively in the text. We will adjust all costs to 2019 pound sterling values using gross domestic product deflators, and use relevant exchange rates for international comparisons.

## Confidence in cumulative evidence

### Quantitative evidence

The quality of evidence will be assessed across the included studies at outcome level for each comparison using GRADE,[57] and incorporated into SoF tables using the GRADEpro software.[58] Our main comparison will be 'shock-absorbing flooring vs standard flooring', and we will include separate SoF tables for hospitals and care homes. Supplementary SoF tables will be developed for different types of shock-absorbing floors versus standard flooring, and for head-to-head comparisons of different shock-absorbing flooring interventions.

The following outcomes will be included: (1) injurious falls rate per 1000 patient-bed days; (2) falls rate per 1000 patient-bed days; (3) number of falls with injuries (eg, none, minor, moderate, severe); (4) number of fractures; (5) number of hip fractures; (6) number of fallers; and (7) number of adverse events related to staff injuries. We will create supporting 'Evidence profile' table.[57] The GRADE system provides a grade of the overall quality of the evidence for each outcome on one of four levels: high, moderate, low, very low.

### Qualitative evidence

The CERQual group's recommendations will be followed to assess the quality of qualitative evidence included in the review.[69] Each review finding will be assessed based on methodological limitations, coherence, adequacy of

data and relevance.[70] We will make an overall assessment of confidence for each review finding on one of the four levels: high, moderate, low, very low. Assessments will be presented in 'CERQual Evidence profile', and 'Summary of Qualitative Findings (SoQF)' tables.

## ENGAGEMENT WITH STAKEHOLDERS

We will consult with key stakeholders and a range of potential knowledge users during our review. Our Advisory Board includes the following knowledge users: Falls in older people National Institute for Health and Care Excellence Guideline Developer; Safety and Improvement Clinical Lead (Leeds Teaching Hospitals NHS Trust); director/chairman of the Health Estates and Facilities Management Association; chairman of the National Care Association; public members; and shock-absorbing flooring researchers from health sciences and engineering disciplines in the UK and Canada. Collectively, members of the board possess the relevant expertise and decision-making authority to critically evaluate and implement shock-absorbing flooring systems in high-risk environments such as hospitals and long-term care in the UK, and use systematic review evidence to inform future research.

An interactive process of communication between researchers and the Advisory Board will be used throughout the review process. We will involve the Board in a number of important ways[1]: in providing input on the design and implementation of the review[2]; as members of the project team who attend project meetings and inform us of emerging primary research evidence[3]; in the interpretation of findings and identification of research gaps; and[4] in the packaging and dissemination of the review's findings in a form that is relevant, practical and easily interpreted by other decision-makers and knowledge users.

## PATIENT AND PUBLIC INVOLVEMENT

Three public members engaged actively in the preparation of our funding proposal. They informed our decisions relating to methodology, particularly prioritising outcomes, confirming settings and development of the theoretical framework.

The public members will participate in five specific patient and public involvement meetings over the course of the project. Each meeting will include a brief training session to explain the stage of the review the project is at, and the processes and tasks involved. They will contribute to the conduct of the systematic review in the following ways: (1) commenting on the clarity and comprehensiveness of the protocol; providing an independent judgement as to the fairness, transparency and consistency of (2) the risk of bias and (3) GRADE judgements made by the project team; (4) providing feedback on the clarity of information presented in the SoF tables, as well as the order and presentation of comparisons and subgroups;

and (5) providing feedback on the clarity, comprehensiveness and presentation of the project outputs (including the plain English summary).

## ETHICS AND DISSEMINATION

We do not need to obtain ethical review, as this is an evidence synthesis. Nonetheless, our ethical considerations[71] will relate to: (1) appropriateness of authorship on the final works; (2) avoidance of duplication in the publication of the findings; (3) avoiding plagiarism by ensuring that all reported findings are sufficiently cited and attributable to the source material; (4) transparency, in the form of acknowledging all contributions and competing interests; (5) having due rigour in the data collection and reporting phases of the review to ensure the accuracy of the findings; and (6) flagging suspected fraudulent or plagiarised research to the publishing journals.

Our research approach is underpinned by the Knowledge to Action Framework,[72] and will ensure involvement of knowledge users with researchers throughout the process.

We will disseminate our research outputs using the following media:
- Open access peer-reviewed journal publication.
- Presentations at national and international conferences, and a webinar.
- Press release/social media with an item in relevant media outlets (eg, The Conversation; The Health Estates and Facilities Management Association 'HEFMA Pulse' magazine).
- A half-day stakeholder symposium, the outputs of which will be made available online.
- A short video distilling the review findings via patient stories.
- Knowledge to Action Reports tailored to NHS chief executives, care home managers and estates/facilities managers, healthcare designers and builders.

**Author affiliations**
[1]School of Health & Care Professions, University of Portsmouth, Portsmouth, UK
[2]School of Engineering, Cardiff University, Cardiff, UK
[3]Department of Kinesiology, University of Waterloo, Waterloo, Ontario, Canada
[4]Department of Biomedical Physiology and Kinesiology, Simon Fraser University, Burnaby, British Columbia, Canada
[5]Faculty of Medicine, University of Southampton, Southampton, UK

**Acknowledgements** We would like to thank our public involvement members, Mrs Margaret Bell, Mrs Elizabeth Burden and Mrs Joleen Tobias, and further members of our Advisory Board: Nadra Ahmed (National Care Association), Alison Cracknell (Leeds Teaching Hospitals NHS Trust), Kirsten Farrell-Savage (University of Portsmouth), Olanrewaju Okunribido (Health & Safety Executive), Jonathan Stewart (Health Estates and Facilities Management Association), Julie Windsor (NHS Improvement) and Anna Winfield (Leeds Teaching Hospitals NHS Trust). Our Advisory Board members have provided feedback and advice throughout the funding application process and when designing the protocol, and have approved the protocol. OO was also a co-applicant on the original funding proposal to HTA.

**Contributors** AD is Principal Investigator (guarantor) of the project, and was involved in the project conception and design, and drafting of the protocol. BK,

CL, AL and DM were all involved in the conception and design of the project, and revising the protocol content. JR contributed to the design of the health economics component and revising the protocol. LF contributed to the design of the Summary of Findings Tables component, and drafting of this manuscript. AD, BK, CL and JR were co-applicants on the original funding proposal to HTA. All authors are also Advisory Board members and have read and approved the manuscript.

**Funding** This report is independent research funded by the National Institute for Health Research (NIHR Health Technology Assessment, 17/148/11 – The SAFEST Review: The Shock-Absorbing Flooring Effectiveness SysTematic Review including older adults and staff in care settings).

**Disclaimer** The views expressed in this publication are those of the authors and not necessarily those of the NHS, the National Institute for Health Research or the Department of Health and Social Care. The Health & Safety Executive are supporting 10 days of Olanrewaju Okunribido's time in kind on this project.

**Competing interests** AD, CL, AL and DM have undertaken studies that will be a part of this review. AD and BK have been collaborating with members of the Health & Safety Executive (2018-present) on some unfunded academic research using a new testing procedure to assess the shock-absorbency of various floor coverings. Five flooring manufacturers delivered free samples to use in the project. AD and BK have no stake in any of these companies. In 2015, AD was involved in a collaborative funding application with Polyflor for some SBRI Healthcare innovation funding. The application was short-listed but unsuccessful. AD has no stake in this company. AL reports grants from SofSurfaces Inc, grants and personal fees from SorbaShock LLC, grants and personal fees from Viconic Sporting, outside the submitted work. AL is a member of an ASTM Work Group (WK38804) whose Technical Contact is the President of SATech. SATech has donated flooring materials to AL's laboratory that have formed the basis of several studies examining the biomechanical effectiveness of compliant flooring (i.e. safety flooring). He has never had (nor does he currently have) any financial links to the company. CL is employed at the Canadian Agency for Drugs and Technologies in Health (CADTH). JR is a member of the NIHR's HTA/EME editorial board (0.1 wte).

**Patient consent for publication** Not required.

**Provenance and peer review** Not commissioned; externally peer reviewed.

**ORCID iDs**
Amy Drahota http://orcid.org/0000-0001-9772-0220
Lambert M Felix http://orcid.org/0000-0001-6517-9089
Bethany E Keenan http://orcid.org/0000-0001-7787-2892
Chantelle C Lachance http://orcid.org/0000-0001-5755-5160
Andrew Laing http://orcid.org/0000-0001-8128-011X
Dawn C Mackey http://orcid.org/0000-0001-9854-1486
James Raftery http://orcid.org/0000-0003-1094-8578

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
