## [Reviewer comments · BMJ Open]

ARTICLE DETAILS

TITLE (PROVISIONAL)	Protocol for The SAFEST Review: The Shock-Absorbing Flooring Effectiveness Systematic Review including older adults and staff in hospitals and care homes
AUTHORS	Drahota, Amy; Felix, Lambert; Keenan, Bethany; Lachance, Chantelle; Laing, Andrew Laing; Mackey, Dawn; Raftery, James

VERSION 1 – REVIEW

REVIEWER	Finn Nilson The Centre for Public Safety and The Department of Risk and Environmental Studies, Karlstad University, Sweden. I have published a number of studies on the proposed subject that will be included in the proposed review. However, I do not see this as a problematic issue.
REVIEW RETURNED	01-Jul-2019

GENERAL COMMENTS	Thank you for the opportunity to review this Study Protocol. This is an important proposed study and generally the study protocol seems well designed. My only concern is regarding the outcomes. Firstly, given that the important outcome is the differentiation of risk, I don't see why having the number of fallers, number of fallers with injuries, etc., is relevant. Secondly, my experience is that there could be a change in the injury panorama as a consequence of using shock-absorbing flooring. I.e. given the reduction in force, a major injury becomes a moderate injury and a moderate injury becomes a minor injury. Therefore, an increase in minor injuries could be positive. Consequentially, I think the authors should differentiate in severity as a specific primary outcome. Thirdly, whilst injury or falls per patient-bed days is often used, it would be preferable if the most important outcome is the risk of injury per fall given that this is the actual underlying purpose of the intervention. Of course, it would be even more beneficial if the injury per fall was controlled by the level of mobility, i.e. the number of meters travelled. My experience is that staff on a ward with shock-absorbing flooring will allow for more mobility, thereby increasing the risk of falls (although it could of course be argued that increased mobility increases strength and thereby decreases the risk of fall). Regardless, these aspects need to be discussed in the outcome measures.
--

REVIEWER	HC Hanger Canterbury District Health Board and University of Otago Christchurch New Zealand
-----------------	--

	I have been an investigator of 2 studies involving shock absorbing flooring which may be included in this systematic review. These are (1) Hanger HC. Low Impact Flooring: Does it reduce fall related injuries? J Am Med Dir Assoc 2017 ;18: 588-591. http://www.sciencedirect.com.cmezproxy.chmeds.ac.nz/science/article/pii/S1525861017300592 and (2) Losco E, Hanger HC, Wilkinson TJ. Ease of walking on low-impact flooring in frail older people. J Am Med Dir Assoc 2019;20(3):385-386 https://www.sciencedirect.com/science/article/pii/S1525861018307151?dgcid=coauthor I do not have any financial interest in any flooring companies and I have never received any personal grants for the above studies
REVIEW RETURNED	17-Jul-2019

GENERAL COMMENTS	Title suggests the review might only include flooring in (residential) care settings, whereas text makes it clear that the review includes hospital settings as well. I would suggest amending title to include ".....in hospital and care settings." Search strategy seems appropriate and extensive. Whilst data extraction is to be managed by 2 independent reviewers, it is not overt that the selection of publications is subject to the same level of independent review. It is implied on page 9, line 13 where a 3rd independent person will adjudicate where conflicting opinion occurs. This implies 2 reviewers but this needs stating overtly. Under potential outcomes, head injuries are not mentioned, but only fractures. Many older people sustain significant head knocks during a fall, with possible traumatic brain injuries. In the literature, these tend to be underestimated as a consequence of falls and should be considered in this review. It is unclear to the reviewer whether wooden floors or subfloors are considered as shock absorbing floors or as controls. p12 suggests wooden subflooring may be shock absorbing flooring, whereas on p6 (interventions and comparators) wooden flooring is not overtly mentioned. They should be explicitly included in one or other group. The manuscript is written in 1st person (we), which I am not used to seeing in scientific publications and I favour writing in 3rd person (it). This is clearly an editorial decision, outside of my scope.
--

REVIEWER	Professor Meg Morris La Trobe University Australia
REVIEW RETURNED	11-Aug-2019

GENERAL COMMENTS	This protocol paper is on an important topic, whether there is strong evidence for the use of shock-absorbing floors for the mitigation of falls in hospitals and residential aged care. The key aims have a high degree of clinical relevance and the topic is applicable globally. The research team are known to be experts in this field. The manuscript would be enhanced by addressing the following issues: 1. The introduction needs to be re-written as it is based mainly on out-dated UK data and does not incorporate recent reports from a wide range of international sources and recent Cochrane reviews. It
--

does not adequately address the wide range of key risk factors for falls in hospitals and aged care and does not adequately discuss the evidence for all of the different environmental risk factors for falls in these settings. The sentence about hip protectors cites an old study and is not linked in to the line of argument. The recent Cochrane review by Cameron et al. particularly needs further discussion in the revised introduction.

2. The writing style needs revision to improve written expression and to remove the over-use of the word "we".

3. The term "care-settings" needs to be clearly operationally defined.

4. The definition of "older adults" needs to be given and justified. It is not adequate to say " We have no set cut-off criteria for age, as chronological age may not be a good indicator of frailty". Please give values.

5. The inclusion criteria for different floors is not clear. Can a study be included if it has the usual hospital floor? Or do special floors such as carpet, low-impact vinyl, cork etc have to be included?

6. Why are studies excluded if people have a mat on the floor? Please explain how mats are not part of the flooring or change this criteria.

7. Please justify in the manuscript why the main comparison is between concrete floors and standard floors? Are concrete floors used very often world-wide? Or change the wording of this section for clarity and generalisability world wide. How will carpet be handled in this analysis?

8. It is hard to understand what is meant by: "The reporting of specific outcomes does not form part of our eligibility criteria". Please revise for specificity - surely the studies you analyse will have to include a primary outcome variable?

9. The dates for the search are confusing as it mixes up text about the scoping review. Please just write about the dates for this new systematic review

10. Are additional electronic databases going to be used? & Cochrane, JBI etc?

11. Please revise the following sentence to remove the confusion text about the prior scoping review "We have refined the search strategy of the scoping review to focus on identifying studies of clinical and cost-effectiveness, and qualitative experiences."

12. The exact strategy for statistical analysis seems to be missing. This needs to be added and referenced.

13. Why is the primary outcome injurious falls per 1000 bed days? Why not falls risk? Please discuss the different ways to measure falls rates and falls risks and accompanying injuries and your decision rules about your selected primary and secondary outcomes.

14. The limitations and generalisability to need to be discussed in great depth, with reference to recent publications from the last 2-5 years.

14. The authors have mentioned that some have industry interests in fall-reduction flooring. AD and BK have been collaborating with members of the Health & Safety Laboratory (2018-present) using a new testing procedure to assess the shock-absorbency of various floor coverings. AD and BK have no stake in any of these companies. In 2015, AD was involved in a collaborative funding application with Polyflor for some SBRIHealthcare innovation funding. Are any of these a conflict of

	interest that needs further discussion?
REVIEWER	Karin Verspoor The University of Melbourne Melbourne, Australia
REVIEW RETURNED	19-Aug-2019
GENERAL COMMENTS	This is a very thorough protocol specification that has carefully followed the PRISMA checklist. In the "Article Summary" I expected to find the bullet point 'Analyses will be at the study level, which limits the scope for exploring moderating factors related to patient-level characteristics on the effectiveness of flooring interventions.' or some statement related to the study scope prior to the more general limitations. The authors indicate that they are building on a scoping review, and therefore exclude any papers prior to May 2016. However, this choice of timeframe could be justified further; it isn't fully clear that despite the existence of a scoping review (cf. a systematic review) that there won't be important literature prior to that date that is relevant to the systematic review.

VERSION 1 – AUTHOR RESPONSE

Reviewer: 1

Reviewer Name: Finn Nilson

Thank you for the opportunity to review this Study Protocol. This is an important proposed study and generally the study protocol seems well designed. My only concern is regarding the outcomes.

Our response: Thank you for your constructive comments. We will respond to your specific concerns below.

1. Firstly, given that the important outcome is the differentiation of risk, I don't see why having the number of fallers, number of fallers with injuries, etc., is relevant.

Our response: We wish to assess the potential benefits and harms of compliant floors. The aim of compliant flooring use in care settings is to reduce the number or severity of injuries from falls. However, whilst it is not currently known, it can be hypothesised that compliant floors may also increase the risk of more vulnerable people falling over. We feel it is therefore important to summarise both injuries and falls as our primary outcomes. Falls and injuries can be summarised in a multitude of ways, and the chosen methods may vary from study to study. The chosen statistics are complicated by the fact that individuals may fall over more than once, and may injure themselves in multiple ways as part of an individual fall, or across multiple falls. Data may be summarised at the level of the individual (e.g. no. of fallers; no. of people sustaining multiple events), at the level of the fall event (e.g. counts of falls; counts of falls with 1 or more injuries), and/or at the level of injury occurrences (e.g. counts of injuries). Depending on the level used and what is included as the denominator, these data can be expressed as rates or risks, and different summary statistics may potentially lead to the interventions appearing more or less favourable, generating a risk of selective outcome reporting.

There is no core outcome set agreed for flooring interventions specifically, however one does exist for community-based fall-related injury prevention trials (Lamb, 2005), and another for hip protector studies (Cameron 2010). Neither are perfectly suited this review, however the former does endorse a range of summary measures to capture the outcome domain of falls. We are interested in the differentiation of risk, and agree with you that this is an important measure; as described in our analysis plan, we will use risk ratios to describe outcomes which are reported at the level of the

individual (e.g. number of fallers), and rate ratios for outcomes reported as counts of events (e.g. number of falls). Since not all studies will necessarily present the risks, we can collect the raw numbers and calculate these for ourselves where required; this also provides greater transparency of where the numbers have come from and allows for a check of the data. We feel that including a range of potential measures is important, as this will help us to assess the risk of selective outcome reporting, and provide us with the opportunity to triangulate the data across summary statistics when drawing conclusions. We have added the words 'risk of falling ≥ 1 times' against the outcome 'number of fallers' (page 11) to clarify that we are interested in assessing risks.

2. Secondly, my experience is that there could be a change in the injury panorama as a consequence of using shock-absorbing flooring. I.e. given the reduction in force, a major injury becomes a moderate injury and a moderate injury becomes a minor injury. Therefore, an increase in minor injuries could be positive. Consequentially, I think the authors should differentiate in severity as a specific primary outcome.

Our response: Thank you for this consideration. Looking at the breakdown in injury severity is one of our secondary outcomes, so will be included in our review. Our primary outcomes (rate of injuries per 1000 bed days; rate of falls per 1000 bed days) have been selected as we feel these provide the best measure of the overall impact (benefit and risk) of a flooring intervention in a care setting environment, taking into account the level of occupancy and follow-up time, and are familiar metrics for the end-users of our review. First, we will assess if there is an overall difference in injury rates (primary outcome), then we will explore whether there is a more nuanced relationship in terms of potential differences in injury severity (secondary outcome). The latter outcome is likely to be more complex to summarise and explore given the possible differences across studies in terms of how they choose to classify severity. We will also be exploring fractures as an outcome, as this is one of the most severe types of injury, to see if there are any differences in this measure.

We have amended our protocol to provide our rationale for our chosen primary outcomes (page 11): "These measures assess the potential benefits and harms of flooring interventions for patients/residents, accounting for occupancy levels and follow-up time; injurious falls rate additionally accounts for variations to the underlying falls rate, as a pragmatic measure of effectiveness."

3. Thirdly, whilst injury or falls per patient-bed days is often used, it would be preferable if the most important outcome is the risk of injury per fall given that this is the actual underlying purpose of the intervention. Of course, it would be even more beneficial if the injury per fall was controlled by the level of mobility, i.e. the number of meters travelled. My experience is that staff on a ward with shock-absorbing flooring will allow for more mobility, thereby increasing the risk of falls (although it could of course be argued that increased mobility increases strength and thereby decreases the risk of fall). Regardless, these aspects need to be discussed in the outcome measures.

Our response: We agree that the purpose of the intervention is to reduce the number of fall-related injuries (or some may describe this aim as preventing severe falls). We believe the best measure to summarise this is by assessing fall-related injury rate per 1000 patient bed-days, because we feel this provides a better reflection of the effectiveness that a compliant flooring may have in practice. Suppose, for example, that compliant flooring increases the risk of falling, but decreases the risk of injury if you do fall; If we took the percentage of falls resulting in injury as our main outcome measure, it may show a reduction in injuries on the compliant floor, but this may obscure the fact that overall injuries in areas with the compliant floor have increased (or remained constant) because many more people are falling over in the first place. Rate of injurious falls per 1000 patient bed-days will account for this, providing a more complete picture of what is likely to happen with regards to injury occurrences in practice. It examines the impact of the intervention across all of the patients/residents exposed to it, taking into account exposure time, rather than just including those who fell in the

denominator.

We agree with you that the proportion of falls resulting in injury is an important measure to consider, however we feel this is more a measure of efficacy, rather than effectiveness, and hence why we would rather retain it as a secondary outcome. We have noted an error on our part, that we should have specified “no. of falls with injuries” instead of “no. of fallers with injuries” (page 11) – as the latter is less likely to be utilised in primary research, and the former is better describing your point; we have amended this in our protocol. We have promoted this outcome to the top of the list of secondary outcomes (page 11) and also inserted a rationale to better justify our chosen primary outcomes (as above).

The point you raise around increases in mobility is an interesting one. If the findings of this review indicate an increased risk of falls on the more compliant floors, there may be a number of explanatory factors to explore to help elucidate on this, e.g. perhaps staff encourage more mobility as you suggest, or perhaps staff maybe less vigilant towards preventing falls due to a perceived ‘safety net’, or perhaps there are changes in practice as to where the most high risk fallers are placed, or perhaps a more compliant floor may make more vulnerable people more unsteady on their feet, or perhaps a combination of any of these factors may be at work. The qualitative data may help explore some of these issues in more depth, to better inform the implications for practice, however any intervention that appears to increase (even non-injurious) falls would need to be considered with great caution, given that the consequences of falls extend beyond that of physical injuries (i.e. to risk of recurrent falls, psychological factors such as fear of falling, and modified behaviours). This is also why we are keen to retain the rate of injurious falls as a more pragmatic measure of effectiveness for our primary outcome, as opposed to risk of injury per fall, which tends to be more explanatory and more closely aligned to assessing efficacy.

Reviewer: 2

Reviewer Name: HC Hanger

1. Title suggests the review might only include flooring in (residential) care settings, whereas text makes it clear that the review includes hospital settings as well. I would suggest amending title to include ".....in hospital and care settings."

Our response: Thank you for your suggestion; we have amended the title to more specifically describe both ‘hospitals and care homes’. In its broadest sense, ‘care settings’ maybe interpreted to include, for example, GP surgeries, day centres, foster homes, etc., which does not properly capture the focus of our review as set out in our inclusion criteria. Care homes is a term which captures residential care settings with and without nursing care, but would not for example include supported living accommodation. If we were to leave the broader term ‘care settings’ in the title alongside the more specific example of ‘hospitals’ we feel this may lead to further confusion.

2. Search strategy seems appropriate and extensive.

Our response: Thank you.

3. Whilst data extraction is to be managed by 2 independent reviewers, it is not overt that the selection of publications is subject to the same level of independent review. It is implied on page 9, line 13 where a 3rd independent person will adjudicate where conflicting opinion occurs. This implies 2 reviewers but this needs stating overtly.

Our response: Thank you, we have stated under the sub-heading ‘selection process’ (page 10): “We will screen titles, abstracts, and full reports independently in duplicate...”.

4. Under potential outcomes, head injuries are not mentioned, but only fractures. Many older people sustain significant head knocks during a fall, with possible traumatic brain injuries. In the literature, these tend to be underestimated as a consequence of falls and should be considered in this review.

Our response: Thank you for this suggestion. We agree that traumatic brain injuries are an important injury that should not be overlooked; they can have considerable implications associated with mortality and social care (e.g. with subdural haemorrhage). In our initial set of proposed outcomes, we anticipated that these would be captured when we consider all injuries, and when we look at the breakdown of injuries by severity. We are also aware that head injuries may be problematic to diagnose, particularly in patients with advanced dementia, making this outcome more prone to measurement error. Nonetheless, based on your suggestion, we shall attempt to report on this outcome separately where it has been presented in the primary studies, and include it as a secondary outcome (added to page 11).

5. It is unclear to the reviewer whether wooden floors or subfloors are considered as shock absorbing floors or as controls. p12 suggests wooden subflooring may be shock absorbing flooring, whereas on p6 (interventions and comparators) wooden flooring is not overtly mentioned. They should be explicitly included in one or other group.

Our response: Thank you for raising this. In response to this comment and comment 5 from Reviewer 3 below, we have inserted the additional sentence for clarity on page 8: "Studies may compare any combination of flooring systems (subfloors and overlays)." Wooden subfloors are listed under examples of interventions (page 7) as they may be considered more shock absorbent than concrete subfloors. We are interested in comparing different types of floors; wooden floors could be considered in either group as part of any individual comparison, depending on the combination of floors being assessed. We do not wish to mandate which group a particular subfloor should be specified as, as individual studies may have incorporated them into one or both groups. Some studies may choose to compare wooden to concrete subfloors, others may hold the subfloor constant and compare different overlays, others may have compared different combinations of overlay and subfloor materials; we will attempt to summarise the evidence for what is known across all the various floor types.

6. The manuscript is written in 1st person (we), which I am not used to seeing in scientific publications and I favour writing in 3rd person (it). This is clearly an editorial decision, outside of my scope.

Our response: Our preference is to write in the active voice as we feel that it makes the writing more engaging and easier to understand. However, based on your feedback and that of Reviewer 3 (comment 2 below), we have toned down our use of the active voice a bit (most pages in the manuscript detailing the methods of the review now contain amendments of this nature). The author guidelines for BMJ Open do not appear to specify a house style in this regard, and on reading articles from the journal we have noted that some articles are written in the first person, whilst others are not. The BMJ does however ask authors to "please write in a clear, direct, and active style" as part of their house style (<https://www.bmj.com/about-bmj/resources-authors/house-style>), so we assume the BMJ Open are likely also to endorse this style.

Reviewer: 3

Reviewer Name: Professor Meg Morris

This protocol paper is on an important topic, whether there is strong evidence for the use of shock-absorbing floors for the mitigation of falls in hospitals and residential aged care. The key aims have a high degree of clinical relevance and the topic is applicable globally. The research team are known to be experts in this field. The manuscript would be enhanced by addressing the following issues:

Our response: Thank you for your constructive feedback. We have addressed your suggestions in

detail below.

1. The introduction needs to be re-written as it is based mainly on out-dated UK data and does not incorporate recent reports from a wide range of international sources and recent Cochrane reviews. It does not adequately address the wide range of key risk factors for falls in hospitals and aged care and does not adequately discuss the evidence for all of the different environmental risk factors for falls in these settings. The sentence about hip protectors cites an old study and is not linked in to the line of argument. The recent Cochrane review by Cameron et al. particularly needs further discussion in the revised introduction.

Our response: Thank you for your suggestion. We have reviewed the introduction and made some amendments to the citing literature as suggested. Our initial submission was based on an abridged version of our protocol used in our funding application, which was targeted at our UK-based funders and was written towards the start of 2018. Our current revision attempts to draw on more international literature and more recent publications, we have added in examples of risk factors, and clarified our line of argument in reference to hip protectors. In addition we have worked to reduce the introduction in length, given the additional clarifications requested elsewhere in our protocol, in order to keep close to the word limits.

2. The writing style needs revision to improve written expression and to remove the over-use of the word "we".

Our response: Thank you for picking this up. We have reduced our use of the active voice, as described above in response to Reviewer 2.

3. The term "care-settings" needs to be clearly operationally defined.

Our response: We have provided a definition of included settings on page 7. Additionally, we have amended the title to state hospitals and care homes, and removed the broad term 'care settings' from the title, which some may find misleading. Please see our response to the first comment of Reviewer 2 above.

4. The definition of "older adults" needs to be given and justified. It is not adequate to say " We have no set cut-off criteria for age, as chronological age may not be a good indicator of frailty". Please give values.

Our response: Thank you for raising this point. We have attempted to take a pragmatic approach to this issue in our systematic review. Our target population is older adults (albeit we are also interested in staff outcomes, who could be of any adult age), and we anticipate that any primary studies in hospitals and care homes would also have targeted areas which are used by older people, who are more at risk of falls and injuries. We have added some words to clarify this in our protocol (page 7): "The target population for the intervention to potentially benefit is older adults in care settings. We have no set cut-off criteria for age, as chronological age may not be a good indicator of frailty (35). Studies must focus on adult populations to be included; studies focussing solely on children will be excluded. We are also interested in staff outcomes."

We do not wish to exclude studies on the basis that their particular setting starts admitted people of a slightly lower age threshold than an arbitrarily decided value of our own choosing, when the data could remain highly informative for the review. One option could have been for us to establish a rule such as, "to be included, the study participants' mean age score minus one standard deviation must be above a certain value", however we have instead opted for a simpler approach that does not rely on arbitrary thresholds. We will assess whether a study has evaluated an intervention which aims to benefit 'older people' (however the researchers of the primary studies have operationalised this), and

collect and transparently report and summarise the data on age, so that readers of the review can assess the applicability to their own context.

5. The inclusion criteria for different floors is not clear. Can a study be included if it has the usual hospital floor? Or do special floors such as carpet, low-impact vinyl, cork etc have to be included?

Our response: Thank you for raising this; we have clarified this in our protocol (page 8) – “Studies may compare any combination of flooring systems (subfloors and overlays).” In essence, yes, usual hospital floors can be included; studies may compare any combination of flooring systems. For example, in the UK, hospital floors are typically comprised of resilient sheet vinyl, which is often quite thin (e.g. 2mm thick) and often laid over a concrete subfloor. However, some settings may have wooden subfloors, or a low-pile carpet as standard. For a study to be included it needs to have a comparison group which attempts to compare the effect of having a more shock-absorbing floor in situ. So for example, a study may compare a standard hospital floor (e.g. 2mm vinyl) against a more novel thicker floor (such as 8mm vinyl), or it might compare a thin vinyl against a more shock-absorbing carpet. It is likely that the intervention (and control) groups across our studies will vary in terms of how shock-absorbing they actually are (e.g. if they were to be measured mechanically), but our inclusion criteria are quite broad to enable us to compare any variety of combinations of shock-absorbency, because currently there is no guidance around how shock-absorbent a floor should be in practice settings.

6. Why are studies excluded if people have a mat on the floor? Please explain how mats are not part of the flooring or change this criteria.

Our response: Thank you for your comment. We perceive mats to be quite different to our target intervention, as they do not provide universal coverage, they are not a permanent fixture, they may pose a trip hazard or unbalance if a person were to step on to the edge, and they do not come with the same implications for installation and practice as do entire floors. We have elaborated on our reasons for excluding mats in the intervention section on page 7: “We will exclude studies reporting exclusively on mats as they are not permanently affixed to the floor and do not provide universal coverage or protection; mats have different implications for installation and practice and are not the focus of this review.”

7. Please justify in the manuscript why the main comparison is between concrete floors and standard floors? Are concrete floors used very often world-wide? Or change the wording of this section for clarity and generalisability world wide. How will carpet be handled in this analysis?

Our response: This is a useful comment as we think our choice of wording has led to some confusion here. The main comparison is not between standard and concrete floors; in this section (page 8) we were listing examples of control groups against which the various (shock absorbing) interventions may be compared. ‘Standard’ floors and concrete and 2mm vinyl, were all examples of control groups. We have amended the word ‘comparison’ to ‘control’ to make this clearer. We have listed carpet as a potential intervention in the section above (as this may for example be considered more shock absorbent than a thin vinyl). We have elaborated to say that any combination of flooring systems being compared would be of interest to the review, to clarify the generalisability to worldwide contexts.

8. It is hard to understand what is meant by: "The reporting of specific outcomes does not form part of our eligibility criteria". Please revise for specificity - surely the studies you analyse will have to include a primary outcome variable?

Our response: Thank you for raising this. We have adhered to the Cochrane standards for conduct of

systematic reviews. On page 11, in the section 'Outcomes and prioritization,' we have listed the outcomes that we will be analysing from the included studies. To be included in the review, studies need not have reported on any specific outcomes. To be included in a specific analysis within the review, we will require the data from studies on the particular outcome we are reporting. It is possible that studies may selectively report outcomes, or report outcomes in different ways, or not fully report outcomes that are non-significant. As part of our review we will be assessing selective outcome reporting, and whether our data are at risk of bias. If we were to exclude studies based on the reporting of specific outcomes, it would not provide us with a complete picture of the totality of the evidence, and we may overlook the potential for bias in our results. We have revised the manuscript (page 8) to more specifically state: "The reporting of specific outcomes does not form part of our eligibility criteria for studies to be included in this review."

9. The dates for the search are confusing as it mixes up text about the scoping review. Please just write about the dates for this new systematic review

Our response: Thank you for highlighting this. It is important that we explain the scoping review, as the searches undertaken by the scoping review are contributing to current systematic review; the studies from the (broader) scoping review are being assessed for inclusion in this systematic review, and our search is an extension of the scoping review's search. We have attempted to make this clearer in the text (page 8): "To avoid duplication of effort, we will build on the search already conducted in a scoping review (17), which completed its search in May 2016. The clinical (12, 16, 18-26, 38-46) and cost-effectiveness (11, 21, 26, 40-41, 47-63) records identified in the scoping review will be assessed for inclusion in the current review. We will continue the search from May 2016 to present, and will not apply any language restrictions."

10. Are additional electronic databases going to be used? & Cochrane, JBI etc?

Our response: We will not be searching these additional databases as this is a systematic review of primary research studies, rather than an overview of other systematic reviews.

11. Please revise the following sentence to remove the confusion text about the prior scoping review "We have refined the search strategy of the scoping review to focus on identifying studies of clinical and cost-effectiveness, and qualitative experiences."

Our response: Thank you for this suggestion. We have attempted to make this clearer: "We have adapted the broader search strategy of the scoping review (17) to make it more specific to the current study (The SAFEST Review)." (page 9).

12. The exact strategy for statistical analysis seems to be missing. This needs to be added and referenced.

Our response: We have added in some additional elaboration into the data synthesis section (page 13) to explain how the generic inverse variance data type works in RevMan (requiring the entry of the natural logarithms), and to clarify that we will use 95% CI throughout. We have explained the summary statistics we plan to use (under 'measures of treatment effect' – page 12) and explained in the section 'data synthesis' (page 13), that should meta-analysis be viable, we will use a random effects model and the generic inverse variance data type in RevMan (with reference). We have explained how we will organise and group studies (page 13 and 15), how we will assess heterogeneity (page 13) and pre-specified the subgroups we will explore if feasible (page 14). We feel we have satisfied the reporting requirements for systematic review protocols in this regard.

13. Why is the primary outcome injurious falls per 1000 bed days? Why not falls risk? Please discuss

the different ways to measure falls rates and falls risks and accompanying injuries and your decision rules about your selected primary and secondary outcomes.

Our response: Thank you for this query. Please see our points above in response to the first peer reviewer (Q1-Q3) with regards to our prioritisation of outcome measures and choice of statistics. We have inserted more of our rationale into the protocol to justify our choice (page 11). We wish to retain our primary outcomes as originally specified. Following consideration of these peer review comments, which have made us reflect on our prioritisation strategy for outcomes, we have re-ordered the list of secondary outcomes to promote 'number of fractures (risk)' and 'number of hip fractures (risk)' above 'fractures per 1000 patient-bed days (rate)' and 'hip fractures per 1000 patient bed-days (rate)', all within our secondary outcomes. This is because fractures and hip fractures are more commonly reported as risks, they are rarer events and individuals are less likely to experience multiple events (and it is therefore less necessary to analyse these data as counts using rates). The re-ordering of our secondary outcomes in this way has implications for the outcomes that will be presented in our Summary of Findings Tables (as only the first 7 prioritised outcomes will be listed, due to the set format of these tables – see page 16), however all outcomes will still be reported within the body of the review.

14. The limitations and generalisability to need to be discussed in great depth, with reference to recent publications from the last 2-5 years.

Our response: This study is currently at the protocol stage, so we will present a comprehensive discussion of the limitations and generalisability of our review when we report upon the findings, and are able to reflect fully on its quality, included studies/data, and coverage. We have followed the BMJ Open guidelines for authors of protocols, and the reporting guidelines for systematic review protocols (PRISMA-P). In the article summary section on page 4, we have provided the key strengths and limitations of the study in line with the BMJ Open guidelines.

15. The authors have mentioned that some have industry interests in fall-reduction flooring. AD and BK have been collaborating with members of the Health & Safety Laboratory (2018-present) using a new testing procedure to assess the shock-absorbency of various floor coverings. AD and BK have no stake in any of these companies. In 2015, AD was involved in a collaborative funding application with Polyflor for some SBRIHealthcare innovation funding. Are any of these a conflict of interest that needs further discussion?

Our response: We have aimed to list anything that may be perceived by others as potential conflicts of interest. The laboratory work that AD and BK have been collaborating on has been undertaken in our own time, without funding, and has been with the goal of evaluating a laboratory testing procedure, which has negligible bearing on the conduct of the SAFEST Review. It is listed merely because various flooring companies supplied some small free flooring samples for us to use in the research; but we have no stake in any of these companies or any reason to give any of them favourable treatment. The declaration of AD regarding her unsuccessful collaborative funding application with Polyflor demonstrates that she has contacts with individuals in the flooring industry, however she has no stake in any company, and indeed we are not aware of any research involving Polyflor products that might be included in the review. Our goal in this review (and in our academic careers more broadly) is to better determine the potential benefits and harms of flooring interventions in order to better inform practice and future research; we have nothing to personally gain from promoting any particular flooring company or demonstrating significant findings. Indeed the two quantitative studies (a pilot study and fully powered study) that our team members have been involved in have demonstrated non-significant findings.

The most important declaration from our perspective is our involvement with some of the primary

studies to be included in this review. This is a double-edged sword, as our experience as researchers on primary studies in this field has equipped us with the knowledge, expertise, and ambition to conduct this systematic review, however we are aware that our prior involvement in the primary studies could be perceived as problematic for example, when it comes to assessing our own research for quality/risk of bias. We have taken steps to address this, by ensuring that the independent assessments of our studies are conducted by people who were not involved in the original research. All of our assessments will be made available in the final report of the review, and our transparency will enable readers to judge for themselves if we have been fair. Our public involvement members will also be involved in helping us improve the clarity/transparency of our reporting and assessing whether our supporting statements for our risk of bias judgements appear consistent and fair across studies. We are pragmatists who believe that no study is perfect, and even when a study is conducted to the best of the researchers' abilities, it may still be at risk of bias; with this shared understanding, we aim to approach the review as objectively as possible.

Reviewer: 4

Reviewer Name: Karin Verspoor

1. This is a very thorough protocol specification that has carefully followed the PRISMA checklist.

Our response: Thank you for your comment.

2. In the "Article Summary" I expected to find the bullet point 'Analyses will be at the study level, which limits the scope for exploring moderating factors related to patient-level characteristics on the effectiveness of flooring interventions.' or some statement related to the study scope prior to the more general limitations.

Our response: We have re-ordered the bullet points in this section (page 4) so that the highlighted bullet point in your comment now comes before the more general limitations of: "The quality of the evidence will be summarised using the GRADE approach, with the strength of the review's findings limited to the quantity and internal validity of the included studies."

3. The authors indicate that they are building on a scoping review, and therefore exclude any papers prior to May 2016. However, this choice of timeframe could be justified further; it isn't fully clear that despite the existence of a scoping review (cf. a systematic review) that there won't be important literature prior to that date that is relevant to the systematic review.

Our response: Thank you for raising this point, which demonstrates that the manuscript requires clarifying. We have made this clearer in the protocol (page 8). We will not be excluding studies just because they have been published prior to May 2016. The scoping review was broader than this systematic review, and therefore it will have already captured any potentially relevant studies that were published prior to May 2016. In order to reduce duplication of effort, we will be assessing the studies included in the scoping review for potential inclusion in this systematic review; this will mean we do not need to re-do all the searches and screening that has already been undertaken for studies published before May 2016. There is a wealth of important literature of relevance to The SAFEST Review that has been published prior to May 2016 – and luckily for us, the scoping review has already done the work of identifying it.

VERSION 2 – REVIEW

REVIEWER	Finn Nilson Karlstad University, Sweden
REVIEW RETURNED	14-Nov-2019

GENERAL COMMENTS	Thank you for your response to my comments. I sincerely look forward to the results of this review.
---

REVIEWER	Karin Verspoor The University of Melbourne
-----------------	---

REVIEW RETURNED	28-Nov-2019
-------------

GENERAL COMMENTS	The authors have done a good job of addressing reviewer concerns and I have no further comments.
--